# Increasing transparency in machine learning through bootstrap simulation and shapely additive explanations

**Alexander A. Huang**[1,2☯], **Samuel Y. Huang**[1,3☯]*

**1** Department of Statistics and Data Science, Cornell University, Ithaca, New York, United States of America,
**2** Department of MD Education, Northwestern University Feinberg School of Medicine, Chicago, Illinois,
United States of America, **3** Department of Internal Medicine, Virginia Commonwealth University School of
Medicine, Richmond, Virginia, United States of America

☯ These authors contributed equally to this work.
* huangs8@vcu.edu

**Data Availability Statement:** All relevant data are within the manuscript and its Supporting information files.

## Abstract

Machine learning methods are widely used within the medical field. However, the reliability and efficacy of these models is difficult to assess, making it difficult for researchers to identify which machine-learning model to apply to their dataset. We assessed whether variance calculations of model metrics (e.g., AUROC, Sensitivity, Specificity) through bootstrap simulation and SHapely Additive exPlanations (SHAP) could increase model transparency and improve model selection. Data from the England National Health Services Heart Disease Prediction Cohort was used. After comparison of model metrics for XGBoost, Random Forest, Artificial Neural Network, and Adaptive Boosting, XGBoost was used as the machine-learning model of choice in this study. Boost-strap simulation (N = 10,000) was used to empirically derive the distribution of model metrics and covariate Gain statistics. SHapely Additive exPlanations (SHAP) to provide explanations to machine-learning output and simulation to evaluate the variance of model accuracy metrics. For the XGBoost modeling method, we observed (through 10,000 completed simulations) that the AUROC ranged from 0.771 to 0.947, a difference of 0.176, the balanced accuracy ranged from 0.688 to 0.894, a 0.205 difference, the sensitivity ranged from 0.632 to 0.939, a 0.307 difference, and the specificity ranged from 0.595 to 0.944, a 0.394 difference. Among 10,000 simulations completed, we observed that the gain for Angina ranged from 0.225 to 0.456, a difference of 0.231, for Cholesterol ranged from 0.148 to 0.326, a difference of 0.178, for maximum heart rate (MaxHR) ranged from 0.081 to 0.200, a range of 0.119, and for Age ranged from 0.059 to 0.157, difference of 0.098. Use of simulations to empirically evaluate the variability of model metrics and explanatory algorithms to observe if covariates match the literature are necessary for increased transparency, reliability, and utility of machine learning methods. These variance statistics, combined with model accuracy statistics can help researchers identify the best model for a given dataset.

**Funding:** The authors received no specific funding for this work.

**Competing interests:** The authors have declared that no competing interests exist.

## Introduction

Machine learning (ML) algorithms generate predictions from sample data without explicit directions from the user [1–4]. Common ML algorithms (e.g., XGBoost, Random Forest, Neural Networks) have been found to be more accurate than traditional parametric methods (linear regression, logistic regression) [5–8]. It has been hypothesized that this increase in accuracy can be attributed to potential non-linear relationships between the independent and dependent variables and interactions between multiple covariates [9, 10]. However, the increase in ML algorithms compared to traditional parametric methods comes at a significant cost: interpretability [11–15]. Linear regression and logistic regression have clear interpretable output that have been widely studied [16–18]. Machine-learning algorithms are often non-interpretable, leading to their reputation as a "black box" algorithm [10, 19–21]. As a result, the interpretability, reliability, and efficacy of machine-learning models is often difficult to assess [14, 20, 22–24].

Without methods that explain how machine learning algorithms reach their predictions, clinicians will not be able to identify if models are reliable and generalizable or just replicating the biases within the training datasets [11, 13, 25]. Provision of explanations about how model predictions are researched and providing accurate summary statistics for model accuracy metrics (e.g., AUROC, Sensitivity, Specificity, F1, Balanced Accuracy) will increase the transparency of machine learning methods and increase confidence when using their predictions [8, 9, 26, 27]. Potential solutions to these weaknesses in machine learning that have been applied within the field of computer science are SHapely Additive exPlanations (SHAP) for model interpretability and bootstrap simulation for quantifying the statistical distribution of model accuracy metrics [28–30]. However, little is known about the efficacy of SHAP and Bootstrap in evaluating machine-learning methods for medical outcomes such as heart disease. Given these limitations in the literature, with data from the England National Health Services Heart Disease Prediction Cohort, we leveraged SHAP to provide explanations to machine-learning output and bootstrap simulation to evaluate the variance of model accuracy metrics.

## Methods

A retrospective, cohort study using the publicly available Heart Disease Prediction cohort (from the England National Health Services database) was conducted. All methods in this research were carried out in accordance with ethical guidelines detailed by the Data Alliance Partnership Board (DAPB) approved national information standards and data collections for use in health and adult social care. The above was approved by the UK Research Ethics Committee (REC). All participants provided written informed consent and their confidentiality was maintained throughout the study.

### Independent variables

Demographic covariates of age and sex were collected. Clinical covariates of Resting blood pressure, fasting blood sugar, cholesterol, resting electrocardiogram (ECG), presence of Angina, and maximum heart rate were collected.

### Dependent variable

The dependent variable of interest was heart disease, as diagnosed by a clinician.

## Model construction and statistical analysis

Descriptive statistics for all patients, patients with heart disease, and patients without heart disease were computed for all covariates and compared using chi-squared tests for categorical variables and t-tests for continuous variables.

Multiple machine-learning methods were evaluated throughout this study (XGBoost, Random Forest, Artificial Neural Network, and Adaptive Boosting). The model metrics were the Area under the Receiver Operator Characteristic Curve (AUROC), Sensitivity, Specificity, Positive Predictive Value, Negative Predictive Value, F1, Accuracy, and Balanced Accuracy. Additionally, the distribution of the Gain statistic, a measure of the percentage contribution of the variable to the model, for each covariate was assessed.

Boost-strap simulation (N = 10,000 simulations) was carried out by varying the train and test sets (70:30), rerunning the model, and assessing model metrics on the test-set. The model metrics from 10,000 simulations were used to construct the distribution for all model metrics and the gain-statistic for all independent covariates. The distribution of each of statistics was evaluated visually through histograms, and analytically through summary statistics (minimum, 5th percentile, 25th percentile, 50th percentile, 75th percentile, 95th percentile, maximum, mean, standard deviation) and the Anderson-Darling test.

The model chosen with best performance would be based upon the median for the distribution of model metrics, not just based upon a singular value (which is what is commonly used in the literature). The model with the highest overall model accuracy would be used to visualize covariates through Shapely Additive Explanations (SHAP). For model explanation, SHAP visualizations were performed for each independent covariate and visualized in figures. These visualizations were evaluated through clinician judgement to evaluate their concordance with understood relationships in cardiology to validate the predictions from the model.

Overall methodology framework is described in Fig 1.

## Results

Of the 918 patients within the cohort, the mean age was 53.51 (SD = 9.43), with 193 females (21%) and 725 males (79%). The mean Resting Blood Pressure was 132.4 (SD = 19.51), cholesterol was 198.8 (SD = 109.38), 214 (23%) of patients had elevated blood sugar, 188 (20%) of patients had Left Ventricular Hypertrophy (LVH), and 178 (19%) had ST-elevation. The mean heart rate was 136.81 (SD = 25.46) and 371 (40%) patients had Angina. Full demographic information listed in Table 1.

Compared to patients without heart disease, patients with heart disease have a greater number of males (90% vs 65%, p<0.01), a higher resting blood pressure (134.2 vs 130.2, p<0.01), increased rates of elevated blood sugar (33% vs 11%, p<0.01), increased rates of ST elevation on ECG (23% vs 15%, p<0.01), and increased Angina (62% vs 13%, p<0.01).

### Overall performance and variability of the models

Full statistics for model metrics in Table 2. The XGBoost model was observed as the most optimal model for this dataset, with the highest median of all model metrics. We observed that the XGBoost models had strong performance, with median AUROC = 0.87, Balanced Accuracy = 0.79, sensitivity = 0.786, and specificity = 0.785. Among 10,000 simulations completed, we observed that the AUROC ranged from 0.771 to 0.947, a difference of 0.176, the balanced accuracy ranged from 0.688 to 0.894, a 0.205 difference, the sensitivity ranged from 0.632 to 0.939, a 0.307 difference, and the specificity ranged from 0.595 to 0.944, a 0.394 difference.

Full statistics for model covariate gain statistics in Table 3. We observe that Angina, Cholesterol, Maximum Heart Rate (MaxHR) and age are the most important predictors within the

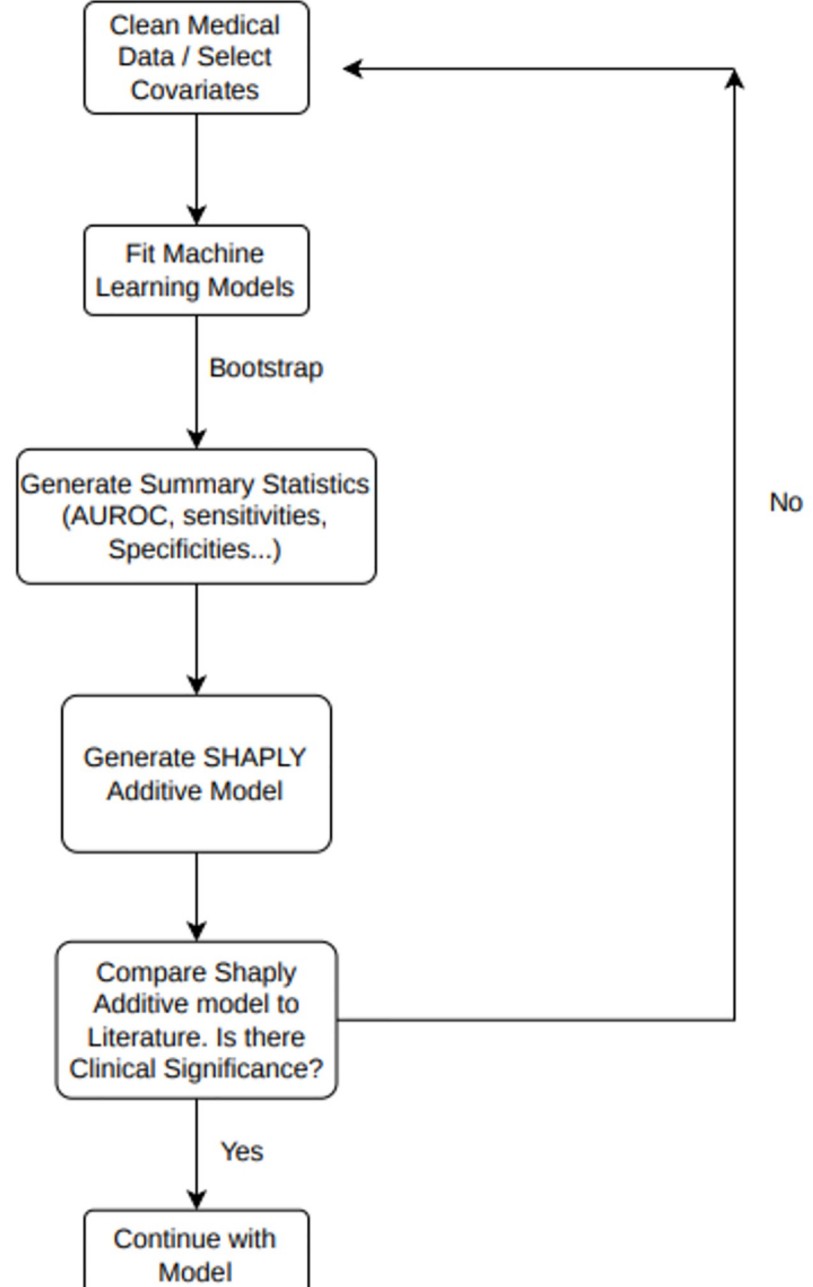

**Fig 1. Consort flow diagram of machine learning workflow.** Description of the overall workflow for machine-learning implementation described within study, starting with a cleaned dataset and ending with a final usable model after critical evaluation of model metrics and visualization of the model through SHAP.

model by the model gain metric. Among 10,000 simulations completed, we observed that the gain for Angina ranged from 0.225 to 0.456, a difference of 0.231, for Cholesterol ranged from 0.148 to 0.326, a difference of 0.178, the MaxHR ranged from 0.081 to 0.200, a range of 0.119, and for Age ranged from 0.059 to 0.157, difference of 0.098.

SHAP analysis was completed and visualized for Angina, Sex, and Max Heart Rate in Fig 2. We observe from SHAP that patients who have Angina, who are of Male gender, and with

**Table 1. Summary of cohort demographics and disease characteristics.**

| Heart Disease Category | | All Patients | No-Heart Disease | Heart Disease | P Values |
|---|---|---|---|---|---|
| Patient Count | | 918 (100%) | 410 (45%) | 508 (55%) | |
| Age | | 53.51 (SD = 9.43) | 50.55 (SD = 9.44) | 55.9 (SD = 9.73) | p<0.01 |
| Gender | Female | 192 (21%) | 143 (35%) | 50 (10%) | p<0.01 |
| | Male | 725 (79%) | 267 (65%) | 458 (90%) | p<0.01 |
| Resting Blood Pressure | | 132.4 (SD = 18.51) | 130.18 (SD = 16.5) | 134.19 (SD = 19.83) | p<0.01 |
| Cholesterol | | 198.8 (SD = 109.38) | 227.12 (SD = 74.63) | 175.94 (SD = 126.39) | p<0.01 |
| Fasting Blood Sugar | Elevated | 214 (23%) | 44 (11%) | 170 (33%) | p<0.01 |
| | Not Elevated | 704 (77%) | 366 (89%) | 338 (67%) | p<0.01 |
| Electrocardiogram | LVH[a] | 188 (20%) | 82 (20%) | 106 (21%) | p<0.01 |
| | Normal | 552 (60%) | 267 (65%) | 275 (56%) | p<0.01 |
| | ST elevation | 178 (19%) | 61 (15%) | 117 (23%) | p<0.01 |
| Maximum Heart Rate | | 136.81 (SD = 25.46) | 148.15 (SD = 23.29) | 127.66 (SD = 23.39) | p<0.01 |
| Angina | No | 547 (60%) | 355 (87%) | 192 (38%) | p<0.01 |
| | Yes | 371 (40%) | 55 (13%) | 316 (62%) | p<0.01 |

[a]LVH = Left Ventricular Hypertrophy

lower maximum heart rates have greater incidence of heart disease, which is concordant with the t-test/chi-squared comparisons that were completed in the Table 1 analysis. All covariates visualized in S1–S5 Figs.

The distributions for all model statistics and the gain statistics for all covariates are in Figs 3 and 4, respectively. The distributions for all model statistics and gain statistics were not significantly different from a normal distribution as ascertained through by the Anderson-Darling Test, using significance of p<0.05 (Table 4).

## Discussion

The use of bootstrap simulation generates 10,000 training and test-set combinations and thus also 10,000 model accuracy statistics and covariate gain statistics [31–33]. This method allows for empiric evaluation of the variability in model accuracy to increase the transparency of model efficacy [34–36].

Prior studies have found that machine learning can be an effective tool to predict outcomes in the medical field such as heart failure, postoperative complications, and infection [15, 37–41]. Shi et al. performed the sequence of fitting ML models and utilized SHAP to determine feature importance to predict postoperative malnutrition in children with congenital heart disease and similarly found XGBoost to provide the most accurate predictions [38]. In a separate study, Lu et al. pulled EHR data from UPMC and found XGBoost could predict EF score [15]. Zhou et. Al utilized a similar paradigm of first comparing machine learning models and then utilizing SHAP for model explanation [39].

What our study brings to the literature is a comprehensive framework for machine learning for medical applications. They consist of an initial machine learning selection methodology that utilizes bootstrap simulation to compute confidence intervals of numerous model accuracy statistics, which is not readily done by current studies. Furthermore, this methodology incorporates multiple feature importance statistics for feature selection. Lastly, the clinically relevant features within the model can be visualized accurately using SHAP. This methodology will streamline the reporting of machine learning by first highlighting the variability of machine

**Table 2. Summary of model metrics for four machine-learning techniques.**

**XGBoost**

| Metrics | Minimum | 5th Percentile | 25th Percentile | Median | 75th Percentile | 95th Percentile | Maximum | Mean | Standard Deviation | Range |
|---|---|---|---|---|---|---|---|---|---|---|
| Accuracy | 0.688 | 0.744 | 0.771 | 0.79 | 0.808 | 0.832 | 0.894 | 0.789 | 0.027 | 0.206 |
| F1 | 0.69 | 0.745 | 0.772 | 0.788 | 0.81 | 0.832 | 0.897 | 0.79 | 0.027 | 0.207 |
| Sensitivity | 0.678 | 0.759 | 0.788 | 0.808 | 0.825 | 0.85 | 0.906 | 0.806 | 0.028 | 0.228 |
| Specificity | 0.595 | 0.709 | 0.753 | 0.785 | 0.814 | 0.855 | 0.944 | 0.784 | 0.042 | 0.349 |
| PPV | 0.68 | 0.757 | 0.786 | 0.82 | 0.845 | 0.88 | 0.954 | 0.82 | 0.037 | 0.274 |
| NPV | 0.57 | 0.678 | 0.725 | 0.756 | 0.787 | 0.83 | 0.928 | 0.756 | 0.046 | 0.358 |
| AUROC | 0.771 | 0.828 | 0.853 | 0.87 | 0.885 | 0.906 | 0.947 | 0.869 | 0.023 | 0.176 |

**Random Forest**

| Metrics | Minimum | 5th Percentile | 25th Percentile | Median | 75th Percentile | 95th Percentile | Maximum | Mean | Standard Deviation | Range |
|---|---|---|---|---|---|---|---|---|---|---|
| Accuracy | 0.670 | 0.728 | 0.768 | 0.782 | 0.800 | 0.815 | 0.889 | 0.784 | 0.026 | 0.219 |
| F1 | 0.683 | 0.736 | 0.772 | 0.781 | 0.806 | 0.815 | 0.880 | 0.786 | 0.026 | 0.196 |
| Sensitivity | 0.663 | 0.747 | 0.784 | 0.797 | 0.807 | 0.846 | 0.893 | 0.797 | 0.029 | 0.229 |
| Specificity | 0.584 | 0.708 | 0.743 | 0.784 | 0.807 | 0.845 | 0.925 | 0.774 | 0.042 | 0.340 |
| PPV | 0.673 | 0.741 | 0.778 | 0.808 | 0.842 | 0.862 | 0.947 | 0.806 | 0.041 | 0.274 |
| NPV | 0.551 | 0.658 | 0.716 | 0.740 | 0.769 | 0.829 | 0.911 | 0.754 | 0.042 | 0.360 |
| AUROC | 0.755 | 0.821 | 0.847 | 0.863 | 0.883 | 0.897 | 0.931 | 0.855 | 0.024 | 0.176 |

**Artificial Neural Network**

| Metrics | Minimum | 5th Percentile | 25th Percentile | Median | 75th Percentile | 95th Percentile | Maximum | Mean | Standard Deviation | Range |
|---|---|---|---|---|---|---|---|---|---|---|
| Accuracy | 0.687 | 0.740 | 0.760 | 0.784 | 0.804 | 0.828 | 0.880 | 0.776 | 0.023 | 0.193 |
| F1 | 0.673 | 0.735 | 0.753 | 0.782 | 0.791 | 0.822 | 0.886 | 0.774 | 0.025 | 0.212 |
| Sensitivity | 0.672 | 0.747 | 0.776 | 0.797 | 0.806 | 0.832 | 0.888 | 0.796 | 0.024 | 0.217 |
| Specificity | 0.594 | 0.704 | 0.751 | 0.769 | 0.799 | 0.837 | 0.926 | 0.764 | 0.039 | 0.332 |
| PPV | 0.660 | 0.749 | 0.778 | 0.811 | 0.836 | 0.862 | 0.939 | 0.808 | 0.033 | 0.278 |
| NPV | 0.551 | 0.662 | 0.715 | 0.748 | 0.771 | 0.814 | 0.913 | 0.744 | 0.050 | 0.362 |
| AUROC | 0.752 | 0.819 | 0.838 | 0.862 | 0.882 | 0.889 | 0.946 | 0.851 | 0.025 | 0.194 |

**Adaptive Boosting**

| Metrics | Minimum | 5th Percentile | 25th Percentile | Median | 75th Percentile | 95th Percentile | Maximum | Mean | Standard Deviation | Range |
|---|---|---|---|---|---|---|---|---|---|---|
| Accuracy | 0.687 | 0.732 | 0.759 | 0.79 | 0.793 | 0.82 | 0.886 | 0.776 | 0.028 | 0.199 |
| F1 | 0.67 | 0.743 | 0.758 | 0.769 | 0.806 | 0.826 | 0.892 | 0.775 | 0.025 | 0.221 |
| Sensitivity | 0.674 | 0.752 | 0.781 | 0.808 | 0.812 | 0.835 | 0.89 | 0.796 | 0.023 | 0.216 |
| Specificity | 0.589 | 0.692 | 0.744 | 0.778 | 0.803 | 0.853 | 0.944 | 0.776 | 0.041 | 0.355 |
| PPV | 0.672 | 0.743 | 0.774 | 0.8 | 0.845 | 0.862 | 0.948 | 0.816 | 0.04 | 0.276 |
| NPV | 0.567 | 0.661 | 0.714 | 0.749 | 0.786 | 0.826 | 0.925 | 0.749 | 0.042 | 0.358 |
| AUROC | 0.756 | 0.814 | 0.839 | 0.865 | 0.865 | 0.897 | 0.934 | 0.866 | 0.026 | 0.178 |

Summary of model metrics within the test set for each of the four machine-learning techniques (XGBoost, Random Forest, Artificial Neural Network, and Adaptive Boosting).

learning accuracy statistics even when utilizing the same dataset, using feature importance statistics to understand how the model values each feature and finally utilizing SHAP visualization to understand how the model is generating predictions from each covariate.

## Overall variability in model accuracy

From simulations, we observed that the AUROC ranged from 0.771 to 0.947, a difference of 0.176. These simulations highlight that for smaller datasets (<10,000 patients), that there may

**Table 3. Summary of model gain statistics for each covariate in the XGBoost model.**

| Covariates | Minimum | 5th Percentile | 25th Percentile | Median | 75th Percentile | 95th Percentile | Maximum | Mean | Standard Deviation | Range |
|---|---|---|---|---|---|---|---|---|---|---|
| Angina | 0.225 | 0.288 | 0.316 | 0.334 | 0.0353 | 0.383 | 0.456 | 0.335 | 0.029 | 0.231 |
| Cholesterol | 0.148 | 0.209 | 0.228 | 0.24 | 0.252 | 0.269 | 0.326 | 0.24 | 0.018 | 0.178 |
| Maximum Heart Rate | 0.081 | 0.114 | 0.129 | 0.139 | 0.15 | 0.165 | 0.201 | 0.139 | 0.015 | 0.12 |
| Age | 0.059 | 0.082 | 0.095 | 0.103 | 0.112 | 0.124 | 0.156 | 0.103 | 0.013 | 0.097 |
| Resting Blood Pressure | 0.027 | 0.051 | 0.061 | 0.069 | 0.076 | 0.087 | 0.109 | 0.069 | 0.011 | 0.082 |
| Sex | 0.026 | 0.038 | 0.044 | 0.049 | 0.054 | 0.062 | 0.082 | 0.049 | 0.007 | 0.056 |
| Fasting Blood Sugar | 0.007 | 0.029 | 0.037 | 0.043 | 0.05 | 0.063 | 0.142 | 0.044 | 0.011 | 0.135 |
| Resting ECG | 0.003 | 0.012 | 0.017 | 0.02 | 0.024 | 0.029 | 0.043 | 0.02 | 0.005 | 0.04 |

be considerable variation in the classification efficacy of the XGBoost model based upon different training-test set combinations [33, 42, 43]. At the higher end, an AUROC of 0.947 implies near perfect fit, while an AUROC of 0.771, while still significantly more predictive than random chance, provides a much decreased level of confidence in the predictions of the model. This highlights a potential issue in replication of machine-learning methods on similar cohorts [22, 44–47]. Two studies may find vastly different results in the predictive accuracy of machine-learning methods even if they use near identical models, covariates, and model summary statistics just due to the choice of the train-test sets (which are determined strictly by random number generation) [32, 35, 36, 48, 49]. As a result, this study highlights the importance of utilizing multiple different train and test sets when executing machine-learning for prediction of clinical outcomes to accurately represent the variance that is present just in the choice of selection of train and test sets [16, 18, 50]. This will accurately characterize the accuracy of the model and allow for better replications of the study. While the only covariate represented in this discussion session is AUROC, these findings were similar within the other accuracy metrics provided in Table 2.

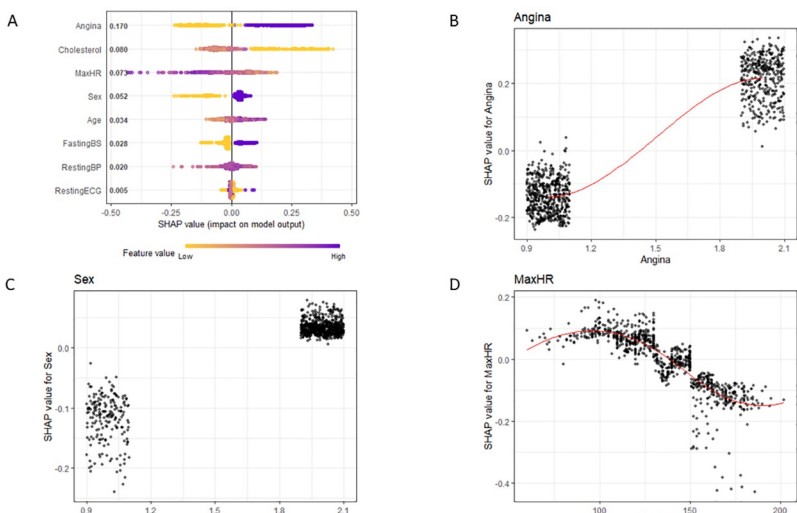

**Fig 2. SHAP analysis.** For the XGBoost models A) Overall Model detailing feature importance, with purple values representing High values and yellow values representing low values of each covariate. B) Model effect for Angina (1 – presence of angina) C) Model effect of Sex (1 –Female, 2 –Male) D) model effect for max heart rate (MaxHR).

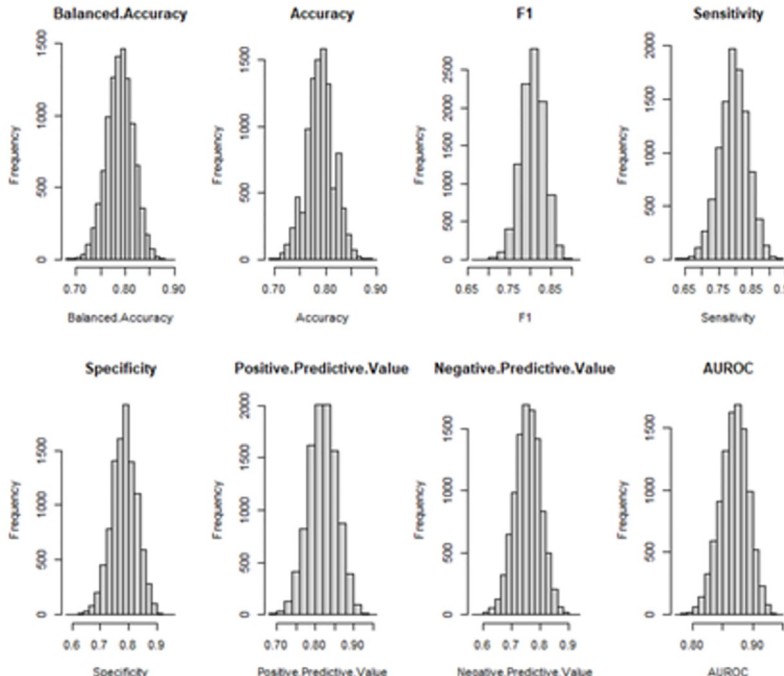

**Fig 3. Model statistics summary.** Balanced accuary, Accuracy, F1, Sensitivty, Specificty, Positive Predictive Value, Negative Predictive Value, Area Under the Receiver Operator Characteristic Curve (AUROC) for the XGboost model following bootstrap simulation.

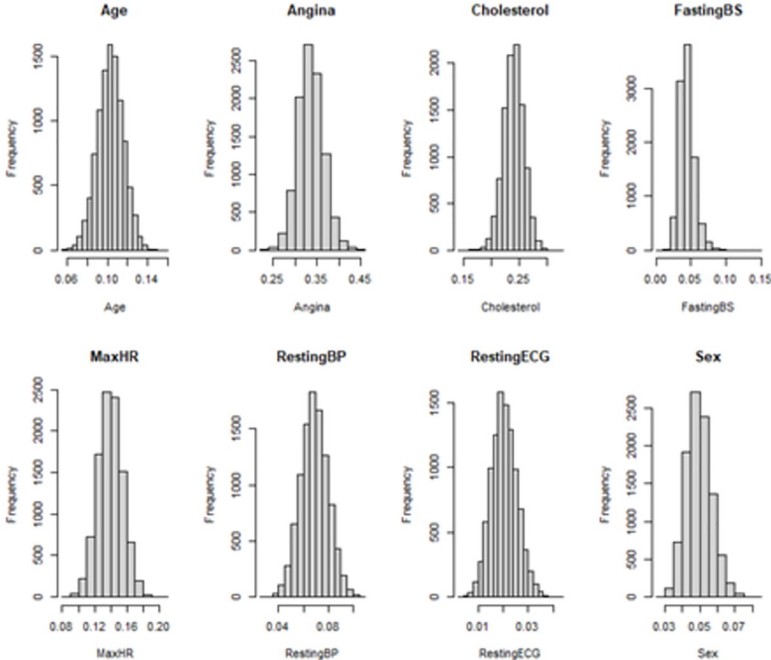

**Fig 4. Gain statistics summary.** For the XGBoost models, the distribution of th gain statistic for all covariates: Age, Angina, Cholesterol, Fasting Blood Sugar (Fasting BS), Maximum Heart Rate (MaxHR), Resting Blood Pressure (RestingBP), Resting electrocardiogram (RestingECG), Sex.

**Table 4. XGBoost model summaries of anderson darling tests.**

| A | | B | |
|---|---|---|---|
| **Model Metrics** | **Anderson Darling P-Value** | **Model Metrics** | **Anderson Darling P-Value** |
| Balanced Accuracy | 0.53 | Balanced Accuracy | 0.23 |
| Accuracy | 0.44 | Accuracy | 0.46 |
| F1 | 0.46 | F1 | 0.3 |
| Sensitivity | 0.18 | Sensitivity | 0.27 |
| Specificity | 0.36 | Specificity | 0.7 |
| Positive Predictive Value | 0.22 | Positive Predictive Value | 0.18 |
| Negative Predictive Value | 0.97 | Negative Predictive Value | 0.99 |
| AUROC | 0.64 | AUROC | 0.1 |

For the XGBoost models: A) Summary of Anderson Darling Test for Normality for Model Metrics B) Summary of Anderson Darling Test for Normality for Gain Statistics for model covariates.

## Overall variability in covariate gain statistics

In addition to capturing the variability in machine-learning methods in model efficacy, there is also significant variability within the gain statistics for each of the covariates. We observed that the gain for Angina ranged from 0.225 to 0.456, a difference of 0.231. Since the gain statistic is a measure of the percentage contribution of the variable to the model, we find that depending on the train and test set, a covariate can have vastly different contributions to the final predictions in the model. This variability in the contribution of each covariate to the final model highlights potential dangers of training-set bias [51, 52]. Depending on which training set is present, a covariate can be twice as important to the final result of the model. This result highlights the need for multiple different "seeds" to be set prior to model training when splitting the training and test sets in order to avoid potential training-set biases and to have the model at least be representative of the cohort it is being trained and tested on (if not representative of the population the cohort is a sample of) [16, 30, 53]. Similar to the model accuracy statistics, this also highlights the difficulty in replication of results in machine-learning models from study to study [1, 54, 55]. Even in our simulation studies with identical cohorts, identical model parameters, and identical covariates, we observed that there was significant variation in which covariates were weighted highly in the final model output. This highlights the need to carefully evaluate the results of the model and not rely on a single seed to set the training and test sets for machine-learning modeling to avoid potential pitfalls that stem from training-test bias [50, 56–61]. While the only covariate represented in this discussion session is Angina, these findings were similar within the other accuracy metrics provided in Table 3.

## Utility of SHAP for model explanation and allowing for augmented intelligence

Given the high level of variability in model accuracy metrics as well as covariate importance based upon different combinations of training and test sets, necessity of algorithms to explain the model are necessary to reduce potential for algorithmic bias. After simulations of model accuracy and covariate gain metrics, a seed can be chosen that accurately represents the center of the distribution for model accuracy metrics and covariate gain statistics. Then SHAP may be executed for Model Explanation to allow for interpretation of model covariates [15, 22, 26].

In traditional parametric methods such as linear regression, each covariate can be interpreted clearly (e.g., for each 1 increase in x, we observe 2 increases in y) [17, 49]. However, due

to the complexity of the non-parametric algorithms that are common in machine-learning methods, it is impossible for a human to analyze each tree and execute an explanation of how the machine-learning method works [1, 62–65]. Thus, using SHAP allows for a similar covariate interpretation as linear regression even if the exact effect-sizes of the covariates cannot be interpreted the way it can in linear regression [15, 22, 49, 66–68]. Fig 2A highlights the relationship between increasing values of a covariate (purple) and increased odds for heart disease. Additionally, Fig 2B–2D allow for observation of the effect sizes of individual covariates. We observe within these plots that patients with Angina lead to significant increase in risk for heart disease, patients who are Male have an increased chance for heart disease, and patients with greater maximum heart rates have a decreased risk for heart disease. In evaluating these three covariates, a researcher/clinician can make judgment calls on if these are concordant with medical literature (prospective clinical trials, retrospective analyses, physiological mechanisms) to validate the results of the model. If the results of the model are not concordant with the medical literature, either a potentially new interpretation of the covariate should be investigated or continued evaluation of if confounders within the model may be done to rectify these observed discrepancies.

## Limitations

This study has several strengths and weaknesses. One weakness is that this study utilizes only one cohort that may not have complete electronic health record data (charts, most labs, diagnoses, or procedural codes) to evaluate model variance. However, since the goal was to evaluate methods to increase transparency in machine-learning instead of developing models for heart disease, this is less of a concern. Furthermore, use of a publicly available dataset already built into an R package allows for increased replicability of this study, which is concordant with the general recommendations within this paper. Another weakness is the need for this methodology to be replicated on other machine-learning methods (neural networks, random-forest) and in other cohorts, both smaller and larger, to get a better understanding of how random chance in selecting training and test sets can significantly impact the perception of model accuracy and the perception of the most important model covariates. Furthermore, this methodology requires a high computational load that would make it difficult to replicate in larger studies with more heterogeneous data. One method to alleviate these issues is pre-selecting covariates that are medically meaningful and have a strong univariable statistical relationship with the outcome. With larger sample sizes, reducing the number of bootstrap simulations can alleviate computational load since a large sample size would naturally decrease variance. Further studies would be needed to utilize this methodology on large heterogeneous electronic health record data.

## Conclusion

Machine learning algorithms are a powerful tool for medical prediction. Use of simulations to empirically evaluate variance of model metrics and explanatory algorithms to observe if covariates match the literature are necessary for increased transparency of machine learning methods, helping to detect true signal in the data instead of perpetuating biases within the training datasets.

## Supporting information

**S1 Fig. SHAP for cholesterol.** Each point represents each observation, the red line represents a trend line. X-axis is the covariate of interest, Cholesterol (mg/day). The SHAP value

represents the log-odds for heart disease.
(TIF)

**S2 Fig. SHAP for age.** Each point represents each observation; the red line represents a trend line. X-axis is the covariate of interest, Age(years). The SHAP value represents the log-odds for heart disease.
(TIF)

**S3 Fig. SHAP for fasting BS.** Each point represents each observation. X-axis is the covariate of interest, Fasting Blood Sugar. Non-elevated = 1, Elevated = 2. The SHAP value represents the log-odds for heart disease.
(TIF)

**S4 Fig. SHAP for resting BS.** Each point represents each observation; the red line represents a trend line. X-axis is the covariate of interest, Resting Blood Pressure (mean arterial pressure). The SHAP value represents the log-odds for heart disease.
(TIF)

**S5 Fig. SHAP for resting ECG.** Each point represents each observation; the red line represents a trend line. X-axis is the covariate of interest, Resting Electrocardiogram. 1 represents an ST-elevation, 2 represents normal, and 3 represents left ventricular hypertrophy. The SHAP value represents the log-odds for heart disease.
(TIF)

**S1 File. Minimal file dataset.** Heart Disease Prediction Cohort from the England National Health Services Database.
(CSV)

## Author Contributions

**Conceptualization:** Alexander A. Huang, Samuel Y. Huang.

**Data curation:** Alexander A. Huang, Samuel Y. Huang.

**Formal analysis:** Alexander A. Huang, Samuel Y. Huang.

**Investigation:** Alexander A. Huang, Samuel Y. Huang.

**Methodology:** Alexander A. Huang, Samuel Y. Huang.

**Project administration:** Alexander A. Huang, Samuel Y. Huang.

**Resources:** Alexander A. Huang.

**Software:** Alexander A. Huang.

**Supervision:** Alexander A. Huang.

**Validation:** Alexander A. Huang.

**Visualization:** Alexander A. Huang.

**Writing – original draft:** Alexander A. Huang, Samuel Y. Huang.

**Writing – review & editing:** Alexander A. Huang, Samuel Y. Huang.

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
