## [Decision Letter · Decision Letter 0]

29 Dec 2022

PONE-D-22-32361Increasing Transparency in Machine Learning through Bootstrap Simulation and Shapely Additive ExplanationsPLOS ONE

Dear Dr. Huang,

Thank you for submitting your manuscript to PLOS ONE. After careful consideration, we feel that it has merit but does not fully meet PLOS ONE’s publication criteria as it currently stands. Therefore, we invite you to submit a revised version of the manuscript that addresses the points raised during the review process. The reviewers' comments are appended below.

We look forward to receiving your revised manuscript.

Kind regards,

Loredana Bellantuono, Ph.D.

Academic Editor

PLOS ONE

Journal Requirements:

2. Thank you for including your ethics statement: "N/A".   

(1) For studies reporting research involving human participants, PLOS ONE requires authors to confirm that this specific study was reviewed and approved by an institutional review board (ethics committee) before the study began. Please provide the specific name of the ethics committee/IRB that approved your study, or explain why you did not seek approval in this case.

(2) Please provide additional details regarding participant consent. In the ethics statement in the Methods and online submission information, please ensure that you have specified (1) whether consent was informed and (2) what type you obtained (for instance, written or verbal, and if verbal, how it was documented and witnessed). If your study included minors, state whether you obtained consent from parents or guardians. If the need for consent was waived by the ethics committee, please include this information.

5. Please note that in order to use the direct billing option the corresponding author must be affiliated with the chosen institute. Please either amend your manuscript to change the affiliation or corresponding author, or email us at plosone@plos.org with a request to remove this option.

6. Please upload a new copy of Figure 1 as the detail is not clear. Please follow the link for more information:

https://blogs.plos.org/plos/2019/06/looking-good-tips-for-creating-your-plos-figures-graphics/

https://blogs.plos.org/plos/2019/06/looking-good-tips-for-creating-your-plos-figures-graphics/

7. Please upload a copy of Supporting Information Figures 1 to 9 which you refer to in your text on page 15.

Reviewers' comments:

Reviewer's Responses to Questions

**Comments to the Author**

1. Is the manuscript technically sound, and do the data support the conclusions?

Reviewer #1: Yes

Reviewer #2: Yes

2. Has the statistical analysis been performed appropriately and rigorously? 

Reviewer #1: Yes

Reviewer #2: Yes

3. Have the authors made all data underlying the findings in their manuscript fully available?

Reviewer #1: Yes

Reviewer #2: Yes

4. Is the manuscript presented in an intelligible fashion and written in standard English?

Reviewer #1: Yes

Reviewer #2: Yes

5. Review Comments to the Author

Reviewer #1: This paper addresses the problem of machine learning explainability to medicine. Through bootstrap simulation and Shapley Additive exPlanations (SHAP), the study aims to increase model transparency and reliability and improve model selection.

Strengths:

1) The paper presented an interesting approach and evaluated the chosen dataset well.

2) The system presented has potential for wide-scale deployment, tested against many strong baselines.

Weaknesses:

Although the paper presented interesting contributions towards the problem of machine learning explainability in health, a few areas still need some improvement.

• Abstract: In the abstract, only the XGBoost was mentioned as the model used, but other models, such as Random Forest, Artificial Neural Network etc was used in the study.

• Compare results with findings in literature: It might be worth comparing the findings of this study with that in the literature. It is essential to know what was learnt and the methodology used.

• SHAP over other tools: Although ML explainability is relatively new, several tools have been explored in the literature. It might be worth asking why the choice of SHAP over others, such as ELI5, LIME and Yellowbrick, etc.

• Sketchy literature: For an emerging study, the literature appears insufficient missing prior work in this area. How is this study different from the below (cite if relevant):

o Dave, D., Naik, H., Singhal, S. and Patel, P., 2020. Explainable ai meets healthcare: A study on heart disease dataset. arXiv preprint arXiv:2011.03195.

o Shi, H., Yang, D., Tang, K., Hu, C., Li, L., Zhang, L., Gong, T. and Cui, Y., 2022. Explainable machine learning model for predicting the occurrence of postoperative malnutrition in children with congenital heart disease. Clinical Nutrition, 41(1), pp.202-210.

o Lu, S., Chen, R., Wei, W., Belovsky, M. and Lu, X., 2021. Understanding Heart Failure Patients EHR Clinical Features via SHAP Interpretation of Tree-Based Machine Learning Model Predictions. In AMIA Annual Symposium Proceedings (Vol. 2021, p. 813). American Medical Informatics Association.

o Zhou, Y., Chen, S., Rao, Z., Yang, D., Liu, X., Dong, N. and Li, F., 2021. Prediction of 1-year mortality after heart transplantation using machine learning approaches: A single-center study from China. International Journal of Cardiology, 339, pp.21-27.

o Chaves, J.M.Z., Chaudhari, A.S., Wentland, A.L., Desai, A.D., Banerjee, I., Boutin, R.D., Maron, D.J., Rodriguez, F., Sandhu, A.T., Jeffrey, R.B. and Rubin, D., 2021. Opportunistic assessment of ischemic heart disease risk using abdominopelvic computed tomography and medical record data: a multimodal explainable artificial intelligence approach. medRxiv.

o Obaido, G., Ogbuokiri, B., Swart, T.G., Ayawei, N., Kasongo, S.M., Aruleba, K., Mienye, I.D., Aruleba, I., Chukwu, W., Osaye, F. and Egbelowo, O.F., 2022. An Interpretable Machine Learning Approach for Hepatitis B Diagnosis. Applied Sciences, 12(21), p.11127.

Reviewer #2: The paper is clearly written, reproducible and technically fine, in terms of statistical analyses performed.

My concern, beyond the limitations that the authors have addressed is to discuss the approach with more complex designs (data cohorts, variables) that the one in use.

The design is the following:

Independent Variables: Demographic covariates (age and sex). Clinical covariates (Resting blood

pressure, fasting blood sugar, cholesterol, resting electrocardiogram (ECG), presence of Angina,

and maximum heart rate).

Dependent variable: heart disease, as diagnosed by a clinician.

Especially from EHR, one expects to find a mix of information sources with data of different nature, and this aspect of multimodality complicates model selection and may require different strategies to measure performance and explanation.

This might calls for more heterogeneous data to be analyzed with methods to be compared, which is in part one of the limitations that the authors have reported.

The authors should discuss more in depth this part.

6. PLOS authors have the option to publish the peer review history of their article (what does this mean?). If published, this will include your full peer review and any attached files.

Reviewer #1: No

Reviewer #2: No

---

## [Author Response · Author response to Decision Letter 0]

30 Jan 2023

Thank you reviewers for listing the documents leading to the correction of the style requirements.

2. Thank you for including your ethics statement: "N/A". 

(1) For studies reporting research involving human participants, PLOS ONE requires authors to confirm that this specific study was reviewed and approved by an institutional review board (ethics committee) before the study began. Please provide the specific name of the ethics committee/IRB that approved your study, or explain why you did not seek approval in this case.

(2) Please provide additional details regarding participant consent. In the ethics statement in the Methods and online submission information, please ensure that you have specified (1) whether consent was informed and (2) what type you obtained (for instance, written or verbal, and if verbal, how it was documented and witnessed). If your study included minors, state whether you obtained consent from parents or guardians. If the need for consent was waived by the ethics committee, please include this information.

Thank you for advising on PLOS ONE requirements for ethical statements. We have updated these statements in the Methods and “ethics statement” field as well as the cover letter.

Thank you for advising on PLOS ONE’s requirements for financial disclosure. We have amended “The Authors received no specific funding for this work” to the cover letter.

Thank you for advising on PLOS ONE’s requirements for data availability. We have attached the minimal dataset and updated the cover letter.

5. Please note that in order to use the direct billing option the corresponding author must be affiliated with the chosen institute. Please either amend your manuscript to change the affiliation or corresponding author, or email us at plosone@plos.org with a request to remove this option. Write we have new affiliation with VCU and will utilize the direct billing option.

Thank you for advising on PLOS ONE’s affiliation with VCU. We have amended the corresponding author to utilize the direct billing option.

6. Please upload a new copy of Figure 1 as the detail is not clear. Please follow the link for more information:

https://blogs.plos.org/plos/2019/06/looking-good-tips-for-creating-your-plos-figures-graphics/

https://blogs.plos.org/plos/2019/06/looking-good-tips-for-creating-your-plos-figures-graphics/

Thank you for bringing this to our attention. We have updated figures with the best resolution.

7. Please upload a copy of Supporting Information Figures 1 to 9 which you refer to in your text on page 15.

Thank you for bringing this to our attention, we have updated supporting figures.

We have added new references to the list as advised by reviewers.

Reviewers' comments:

Reviewer's Responses to Questions

Comments to the Author

1. Is the manuscript technically sound, and do the data support the conclusions?

Reviewer #1: Yes

Reviewer #2: Yes

2. Has the statistical analysis been performed appropriately and rigorously?

Reviewer #1: Yes

Reviewer #2: Yes

3. Have the authors made all data underlying the findings in their manuscript fully available?

Reviewer #1: Yes

Reviewer #2: Yes

4. Is the manuscript presented in an intelligible fashion and written in standard English?

Reviewer #1: Yes

Reviewer #2: Yes

5. Review Comments to the Author

Reviewer #1: This paper addresses the problem of machine learning explainability to medicine. Through bootstrap simulation and Shapley Additive exPlanations (SHAP), the study aims to increase model transparency and reliability and improve model selection.

Strengths:

1) The paper presented an interesting approach and evaluated the chosen dataset well.

2) The system presented has potential for wide-scale deployment, tested against many strong baselines.

Weaknesses:

Although the paper presented interesting contributions towards the problem of machine learning explainability in health, a few areas still need some improvement.

• Abstract: In the abstract, only the XGBoost was mentioned as the model used, but other models, such as Random Forest, Artificial Neural Network etc was used in the study.

Thank you for bringing this to our attention we have made the change.

• Compare results with findings in literature: It might be worth comparing the findings of this study with that in the literature. It is essential to know what was learnt and the methodology used.

Thank you for bringing this to our attention, we have made the change.

• SHAP over other tools: Although ML explainability is relatively new, several tools have been explored in the literature. It might be worth asking why the choice of SHAP over others, such as ELI5, LIME and Yellowbrick, etc. 

We did a literature search and found SHAP to be most prevalent over others. Additionally, SHAP has packages within R and python that are compatible with machine learning methods used in this study. Further research is needed to compare the efficacy of models such as SHAP with ELI5, LIME, and Yellowbrick, etc. Goal of paper is to generate a workflow for machine learning, taking into account bootstrapping to generate the distribution for model accuracy statistics, evaluate the feature importances of all our covariates, and highlight the utility of model explanation, only SHAP was used in the study, but we agree with the suggestions above and will do it as a line of inquiry in our next. 

• Sketchy literature: For an emerging study, the literature appears insufficient missing prior work in this area. How is this study different from the below (cite if relevant):

o Dave, D., Naik, H., Singhal, S. and Patel, P., 2020. Explainable ai meets healthcare: A study on heart disease dataset. arXiv preprint arXiv:2011.03195.

o Shi, H., Yang, D., Tang, K., Hu, C., Li, L., Zhang, L., Gong, T. and Cui, Y., 2022. Explainable machine learning model for predicting the occurrence of postoperative malnutrition in children with congenital heart disease. Clinical Nutrition, 41(1), pp.202-210.

o Lu, S., Chen, R., Wei, W., Belovsky, M. and Lu, X., 2021. Understanding Heart Failure Patients EHR Clinical Features via SHAP Interpretation of Tree-Based Machine Learning Model Predictions. In AMIA Annual Symposium Proceedings (Vol. 2021, p. 813). American Medical Informatics Association.

o Zhou, Y., Chen, S., Rao, Z., Yang, D., Liu, X., Dong, N. and Li, F., 2021. Prediction of 1-year mortality after heart transplantation using machine learning approaches: A single-center study from China. International Journal of Cardiology, 339, pp.21-27.

o Chaves, J.M.Z., Chaudhari, A.S., Wentland, A.L., Desai, A.D., Banerjee, I., Boutin, R.D., Maron, D.J., Rodriguez, F., Sandhu, A.T., Jeffrey, R.B. and Rubin, D., 2021. Opportunistic assessment of ischemic heart disease risk using abdominopelvic computed tomography and medical record data: a multimodal explainable artificial intelligence approach. medRxiv.

o Obaido, G., Ogbuokiri, B., Swart, T.G., Ayawei, N., Kasongo, S.M., Aruleba, K., Mienye, I.D., Aruleba, I., Chukwu, W., Osaye, F. and Egbelowo, O.F., 2022. An Interpretable Machine Learning Approach for Hepatitis B Diagnosis. Applied Sciences, 12(21), p.11127.

Thank you for suggesting these papers. They have been helpful in revising the introduction and discussion to add greater depth. All of these papers are references either in the introduction or the discussion. What our study brings to the literature is a comprehensive framework for machine learning for medical applications. They consist of an initial machine learning selection methodology that utilizes bootstrap simulation to compute confidence intervals of numerous model accuracy statistics, which is not readily done by current studies. Furthermore, this methodology incorporates multiple feature importance statistics for feature selection. Lastly, the clinically relevant features within the model can be visualized accurately using SHAP. This methodology will streamline the reporting of machine learning by first highlighting the variability of machine learning accuracy statistics even when utilizing the same dataset, using feature importance statistics to understand how the model values each feature and finally utilizing SHAP visualization to understand how the model is generating predictions from each covariate.

Reviewer #2: The paper is clearly written, reproducible and technically fine, in terms of statistical analyses performed.

My concern, beyond the limitations that the authors have addressed is to discuss the approach with more complex designs (data cohorts, variables) that the one in use. 

The design is the following:

Independent Variables: Demographic covariates (age and sex). Clinical covariates (Resting blood

pressure, fasting blood sugar, cholesterol, resting electrocardiogram (ECG), presence of Angina,

and maximum heart rate).

Dependent variable: heart disease, as diagnosed by a clinician.

Especially from EHR, one expects to find a mix of information sources with data of different nature, and this aspect of multimodality complicates model selection and may require different strategies to measure performance and explanation.

This might calls for more heterogeneous data to be analyzed with methods to be compared, which is in part one of the limitations that the authors have reported.

The authors should discuss more in depth this part.

The authors would like to thank the reviewer for highlighting such an important further analysis. The authors have addressed this in the limitations and discussed it more in depth.

---

## [Editor Report · Decision Letter 1]

5 Feb 2023

Increasing Transparency in Machine Learning through Bootstrap Simulation and Shapely Additive Explanations

PONE-D-22-32361R1

Dear Dr. Huang,

We’re pleased to inform you that your manuscript has been judged scientifically suitable for publication and will be formally accepted for publication once it meets all outstanding technical requirements.

Kind regards,

Loredana Bellantuono, Ph.D.

Academic Editor

PLOS ONE
---

## [Editor Report · Acceptance letter]

10 Feb 2023

PONE-D-22-32361R1 

Increasing Transparency in Machine Learning through Bootstrap Simulation and Shapely Additive Explanations 

Dear Dr. Huang:

I'm pleased to inform you that your manuscript has been deemed suitable for publication in PLOS ONE. Congratulations! Your manuscript is now with our production department. 

Kind regards, 

on behalf of

Dr. Loredana Bellantuono 

Academic Editor

PLOS ONE